# Abdominal aortic aneurysm and cardiometabolic traits share strong genetic susceptibility to lipid metabolism and inflammation

Shufen Zheng[1,2], Philip S. Tsao [3,4,5] ✉ & Cuiping Pan[1,2] ✉

Abdominal aortic aneurysm has a high heritability and often co-occurs with other cardiometabolic disorders, suggesting shared genetic susceptibility. We investigate this commonality leveraging recent GWAS studies of abdominal aortic aneurysm and 32 cardiometabolic traits. We find significant genetic correlations between abdominal aortic aneurysm and 21 of the cardiometabolic traits investigated, including causal relationships with coronary artery disease, hypertension, lipid traits, and blood pressure. For each trait pair, we identify shared causal variants, genes, and pathways, revealing that cholesterol metabolism and inflammation are shared most prominently. Additionally, we show the tissue and cell type specificity in the shared signals, with strong enrichment across traits in the liver, arteries, adipose tissues, macrophages, adipocytes, and fibroblasts. Finally, we leverage drug-gene databases to identify several lipid-lowering drugs and antioxidants with high potential to treat abdominal aortic aneurysm with comorbidities. Our study provides insight into the shared genetic mechanism between abdominal aortic aneurysm and cardiometabolic traits, and identifies potential targets for pharmacological intervention.

Abdominal aortic aneurysm (AAA), defined as focal dilation of the abdominal aorta by 50% or reaching ≥ 30 mm in diameter, is a complex vascular disease with an estimated global prevalence of 0.92%[1]. It is asymptomatic in early disease stages, with most AAA discovered by incidental imaging or screening protocols. Once reaching 55 mm, the five-year cumulative rupture rate is 25-40%[2]. Among ruptured patients, a mortality rate as high as 80% was observed[3], rendering AAA a leading cause of death.

AAA is characterized by remodeling and degradation of the extracellular matrix, apoptosis of smooth muscle cells, luminal thrombosis, and chronic inflammation[4,5]. Plaques consisting of lipids, blood cells and other plasma substances accumulate around the lesion sites, with abundant infiltration of innate and adaptive immune cells both in the thrombus and the arterial wall[6]. Meanwhile, metabolic homeostasis can be perturbed, resulting in enhanced glycolysis in the aortic wall[7] and altered serum levels of amino acids and lipids[8–10]. Often, circulating total cholesterol, low-density lipoprotein cholesterol (LDL-C), triglycerides, and sulfur amino acids are elevated, whereas high-density lipoprotein cholesterol (HDL-C) and phosphatidylcholines are reduced. These changes resemble numerous other cardiovascular diseases (CVDs), such as coronary artery disease (CAD), myocardial infarction (MI), and peripheral arterial disease[11]. Indeed,

[1]Center for Intelligent Medicine Research, Greater Bay Area Institute of Precision Medicine (Guangzhou), Guangzhou, China. [2]Center for Evolutionary Biology, Intelligent Medicine Institute, School of Life Sciences, Fudan University, Shanghai, China. [3]Division of Cardiovascular Medicine, Stanford University School of Medicine, Stanford, California, USA. [4]Stanford Cardiovascular Institute, Stanford University, California, USA. [5]VA Palo Alto Health Care System, Palo Alto, California, USA. ✉e-mail: ptsao@stanford.edu; pancuiping@ipm-gba.org.cn

atherosclerosis occurs in 25–55% AAA patients[12], and known risk factors of AAA including male sex, age, smoking, hypercholesterolemia, hyperlipidemia, and hypertension[13], are widely shared among CVDs.

AAA is highly heritable, with an estimated 70% heritability by family and twin studies[14,15]. In fact, high heritability is generally observed in cardiometabolic disorders[16,17], rendering genetic studies a valuable tool to decipher the disease mechanisms[18]. Genome-wide association studies (GWAS), particularly those performed in recent years with large sample sizes, have uncovered single nucleotide variants (SNVs) associated with many complex diseases[19]. A recent meta-GWAS of AAA examined 39,221 cases and 1,086,107 controls, resulting in 141 susceptible loci[20], a several-fold increase in disease loci compared to earlier studies[21,22]. Similarly, recent GWAS provided comprehensive variant profiles for dozens of cardiometabolic traits (CMTs), which have greatly enhanced our understanding of these diseases.

In this study, we leverage these large GWAS data to identify genetic factors shared by AAA and CMTs. We aim to identify shared SNVs and genes, as well as the enriched pathways, cell types, and tissues. Importantly, these results offer valuable information for prioritizing drugs that target shared genes for treating AAA with comorbid conditions.

## Results
### GWAS datasets
We obtained GWAS summary statistical data for 18 cardiometabolic diseases (CMDs) including AAA, 15 metabolic traits, and 6 immune cell traits (Fig. 1A). These traits are distributed over a broad spectrum of cardiac and metabolic functions, including heart functions, vascular circulation, glucose metabolism, lipid metabolism, and immunity. Most of the CMDs were studied in more than 10,000 case samples, whereas metabolic traits and immune cells were measured in a minimum of 560,000 individuals. Although European ancestry was dominant, many studies included various ancestral groups. Furthermore, the number of interrogated genotypes ranged between 4.5–52 million, and the significant SNVs ($P < 5 \times 10^{-8}$) were ample (Supplementary Data 1). Overall, these datasets present a state-of-the-art discovery power for common SNVs-based genetic susceptibility to cardiometabolic disorders. Around these datasets, we designed analysis modules to elucidate the shared genetic architecture of AAA and CMTs, including shared SNVs, genes, pathways, tissues, and cell types (Fig. 1B). Coherent signals from various analyses are found and presented below.

### Genetic correlation
Genome-wide correlations computed by LDSC[23] suggest positive correlations between AAA and 20 CMTs (Fig. 2A). The highest correlated traits are aortic aneurysms, followed by numerous diseases including MI, CAD, peripheral artery disease, subarachnoid hemorrhage, and heart failure ($r_g >= 0.3$, $P < 1 \times 10^{-10}$). Compared to the disorders, the physiological traits display weaker correlations, with lipids, adiposity, blood pressure, and glucose traits in descending order. Only HDL-C presented a negative correlation with AAA ($r_g = -0.25$, $P = 7.61 \times 10^{-32}$). Immune cell counts and percentages did not correlate with AAA, and thus were excluded from subsequent analyses. We also computed genetic correlation by functional elements. Repressors, enhancers and promoters tend to have the strongest correlations across traits (Supplementary Fig. 1), suggesting transcriptional regulation is genetically shared.

### Causal Inference
Many cardiometabolic disorders share risk factors, rendering genetic correlation a result of complex pleiotropic effects. Mendelian Randomization (MR) overcomes the confounding factor issue and provides causal inference. We conducted bidirectional MR using several models and found a mutual causality between AAA and CAD (Fig. 2B). Furthermore, AAA was suggested as causal to MI. Reversely, 10 traits were inferred as causal to AAA, including hypertension (OR = 2.01, $P = 3.36 \times 10^{-4}$), lipid and adiposity traits (OR = 1.46-1.73, $P < 1.24 \times 10^{-12}$), CAD (OR = 1.23, $P = 2.34 \times 10^{-5}$), and diastolic blood pressure (OR = 1.05, $P = 1.13 \times 10^{-11}$). Conversely, HDL-C (OR = 0.65, $P = 2.28 \times 10^{-21}$) and pulse pressure (OR = 0.97, $P = 2.65 \times 10^{-8}$) were causally protective against AAA. Note that no apparent horizontal pleiotropy was detected as the intercept of MR-Egger did not significantly deviate from zero (Supplementary Table 1).

### Cross-trait loci and causal variants
Through cross-trait meta-analysis by MTAG (Multi-Trait Analysis of GWAS)[24] and CPASSOC (Cross-Phenotype Association Analysis)[25], we identified 203 SNVs collectively shared by the 21 trait pairs (Supplementary Data 2). Overall, AAA shares the largest number of SNVs with CAD (N = 46), followed by lipid traits (about 20−40 SNVs) (Supplementary Fig. 2). Next, to derive shared causal SNVs, we first fine-mapped the SNVs with FM-summary[26] for a 99% credible set, and then colocalized these SNVs across traits by Coloc[27]. As such, a total of 177 causal variants shared by two traits were derived (Supplementary Data 3). We also applied HyPrColoc[28] and derived 47 causal variants shared by multiple traits (Fig. 3A). Among the 47 shared causal variants, only four had the smallest GWAS $P$ values (Fig. 3B), reinforcing that local lead SNVs in GWAS may only tag the causal SNVs[26].

We observed the shared SNVs, both causal and non-causal, clustered proximal to lipid-related genes (Supplementary Data 3). For example, *LPA* was the closest gene for 9 SNVs shared by AAA and 11 other traits, among which rs10455872[29] was causal to 4 trait pairs, and rs140570886[30], rs76735376, and rs6905073 were shared by at least 3 trait pairs. Similarly, CDKN2B-AS1 was annotated to 8 SNVs shared by 10 trait pairs, including rs1537371[31] which was causal to 3 trait pairs. We also rediscovered rs12740374[32] on *CELSR2* and rs11591147[33] on *PCSK9*. Lastly, several shared causal SNVs were proximal to *CETP*, *BUD13*, *TRIB1*, *LPL*, and *APOE*, all of which encode lipid regulators and have been associated with CMDs[34–38].

### Shared genes and pathways
Annotating GWAS variants to genes solely by proximity is oversimplified and may not account for pleiotropy. We therefore adopted four approaches, TWAS-Fusion[39], SMR[40], MAGMA[41], and GCTA-fastBAT[42] to infer shared genes (Supplementary Fig. 3). Among these methods, the first two leverage expression quantitative trait loci (eQTL), and the latter two mainly utilize proximity for gene burden tests. We define disease genes as reported by all four methods and thus derived 405 genes (Supplementary Data 4), of which 109 genes were linked to minimally three AAA-trait pairs (Supplementary Fig. 4). Notably, *CELSR2*, *PSRC1*, *LRP1*, and *NOC3L* were each shared among 14 AAA-trait pairs or more. Such broad distribution suggests their essential roles in cardiac and metabolic functions. Interestingly, all four genes participate in lipid metabolism; furthermore, all but *NOC3L* have been reported in inflammation[43,44].

Pooling genes from any of the four methods for an overview of biological pathways, we discovered that their functions were enriched in lipoprotein organization, cholesterol transport, and acylglycerol homeostasis (Supplementary Fig. 5A). Strikingly, cholesterol metabolism was the most enriched pathway across all 21 trait pairs (Supplementary Fig. 5B). When classifying by etiological mechanisms[20], the most prominent enrichments appeared in cholesterol metabolism, PPAR pathway in lipid metabolism, TGF-β pathway in inflammation, and ECM-receptor interaction in extracellular matrix dysregulation (Fig. 3C).

Summarizing the shared SNVs and genes, we construct the comorbidity network for AAA, detailing the shared variants and genes for each trait pair (Fig. 4).

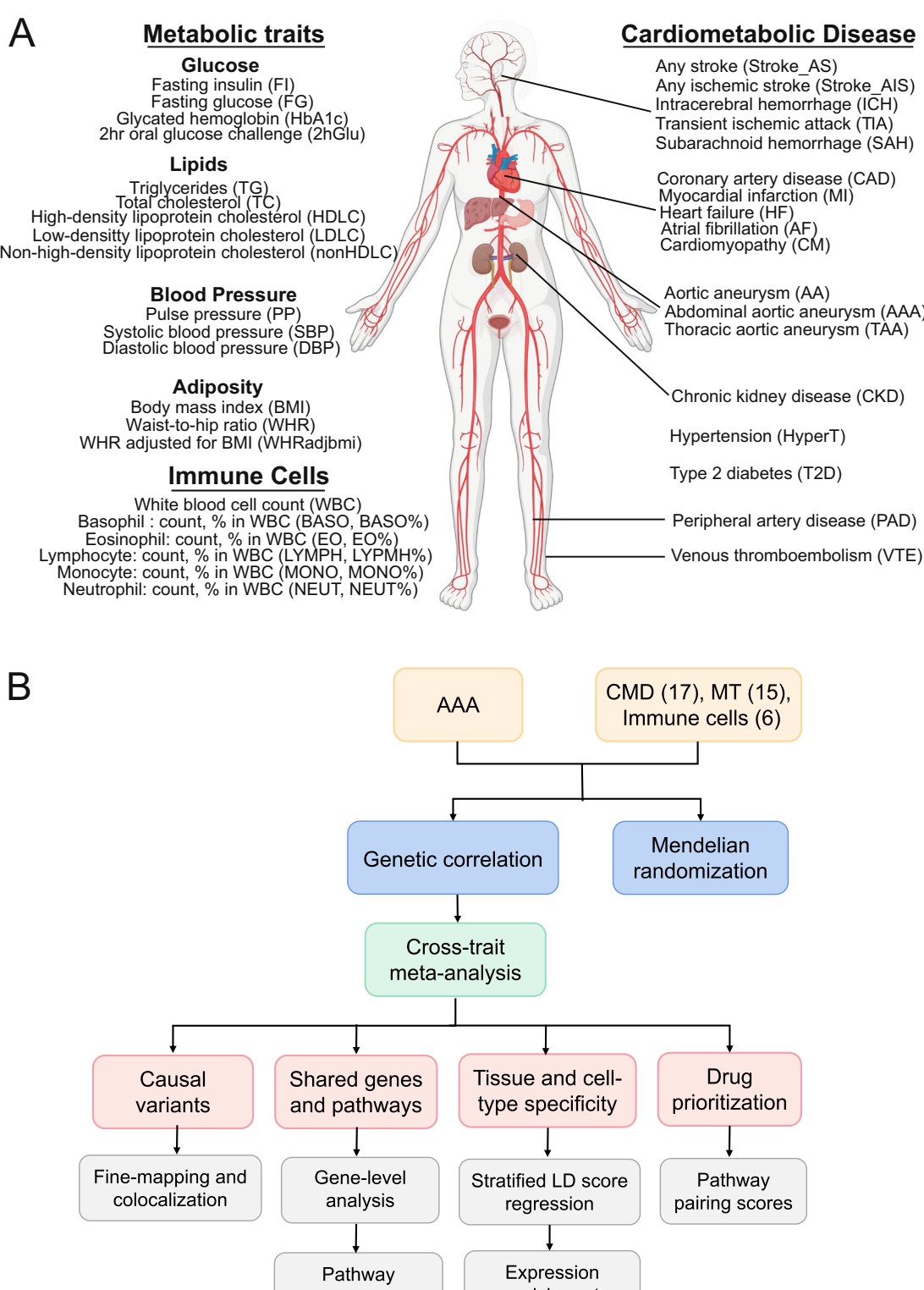

**Fig. 1 | Interrogation of genetic components between AAA and the traits related to cardiometabolism. A** Traits and diseases in this study include 18 cardiometabolic diseases, 15 metabolic traits, and 6 immune cell traits. This graph was created via https://www.biorender.com/. **B** Analysis modules included computing genome-wide genetic correlations, inferring causality between AAA and the traits by bidirectional Mendelian randomization, identifying shared causal variants, genes and pathways, discovering tissues and cell being impacted the most by the shared signals, and prioritizing drugs for treating AAA comorbidities. CMD: cardiometabolic diseases, MT: metabolic traits.

## Tissue and cell-type specificity
The shared genes may function in certain tissues and cell types more specifically. We examined it from gene expression in GTEx[45] and single-cell transcriptome, as well as heritability in tissue-specific genes and cell type-specific enhancers in CATLAS[46]. Combing both approaches, we discovered that liver, artery, and adipose tissue (Supplementary Fig. 6), and adipocytes, hepatocytes, fibroblasts, vascular smooth muscle cells, macrophages, and myeloid cells

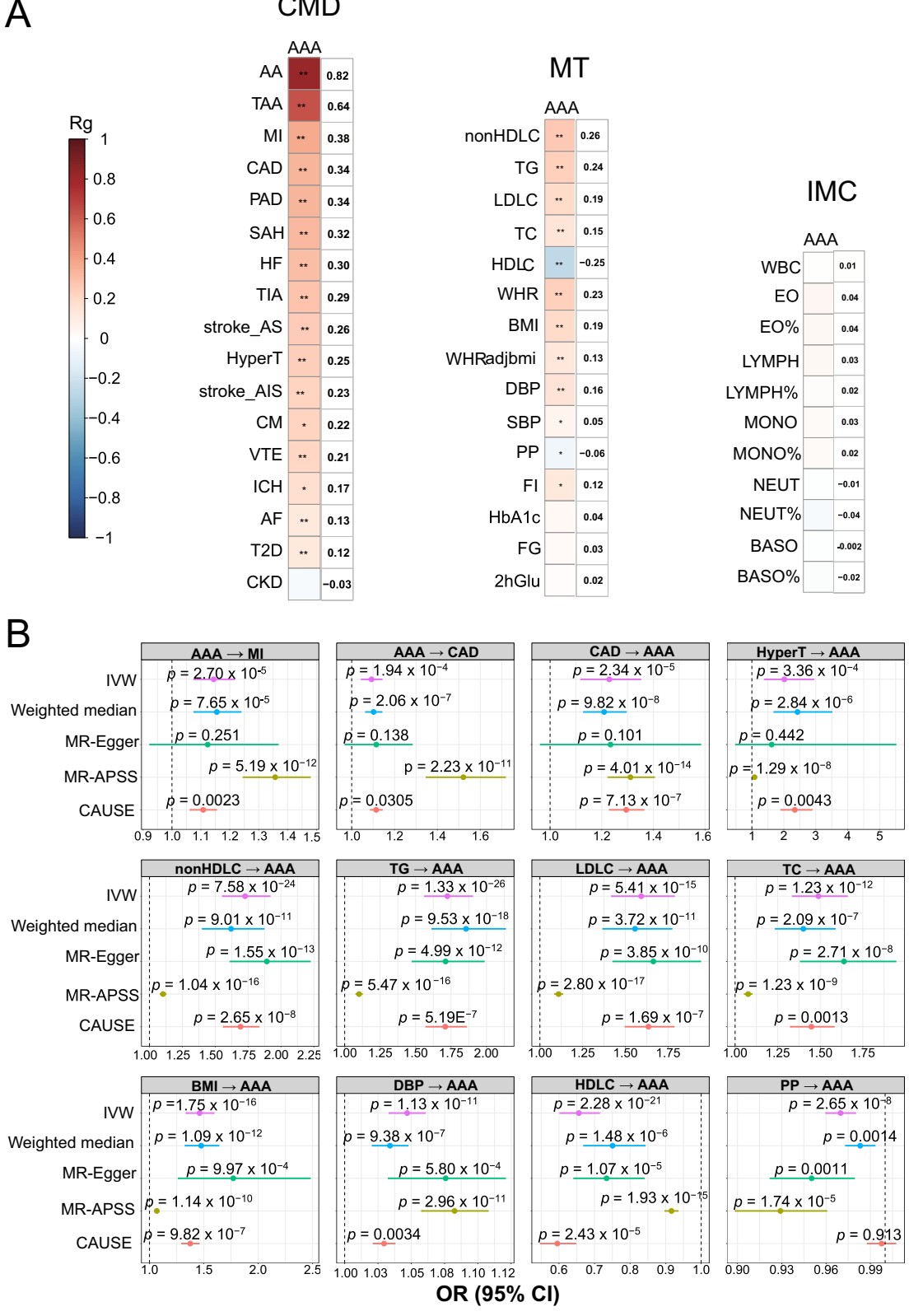

**Fig. 2 | Genetic correlation and causal inference between AAA and CMTs. A** The heatmap presents the genetic correlation $r_g$ calculated in LDSC, with the color scale indicating the strength of the correlation, and the $r_g$ value displayed next to the heatmap. The * marks the statistical significance: *: $P < 0.05$; **: $P < 0.0016$ (Bonferroni-corrected $P$ value threshold). **B** Causal inference by two-sample Mendelian Randomization with five methods. Odds ratios are shown as dots, the color bars present +/− 95% confidence intervals, and $P$ values are depicted above the bars. CMD: cardiometabolic diseases, MT: metabolic traits. IMC: immune cell traits. All reported $P$ values are two-sided, unless stated otherwise. Source data are provided with this paper.

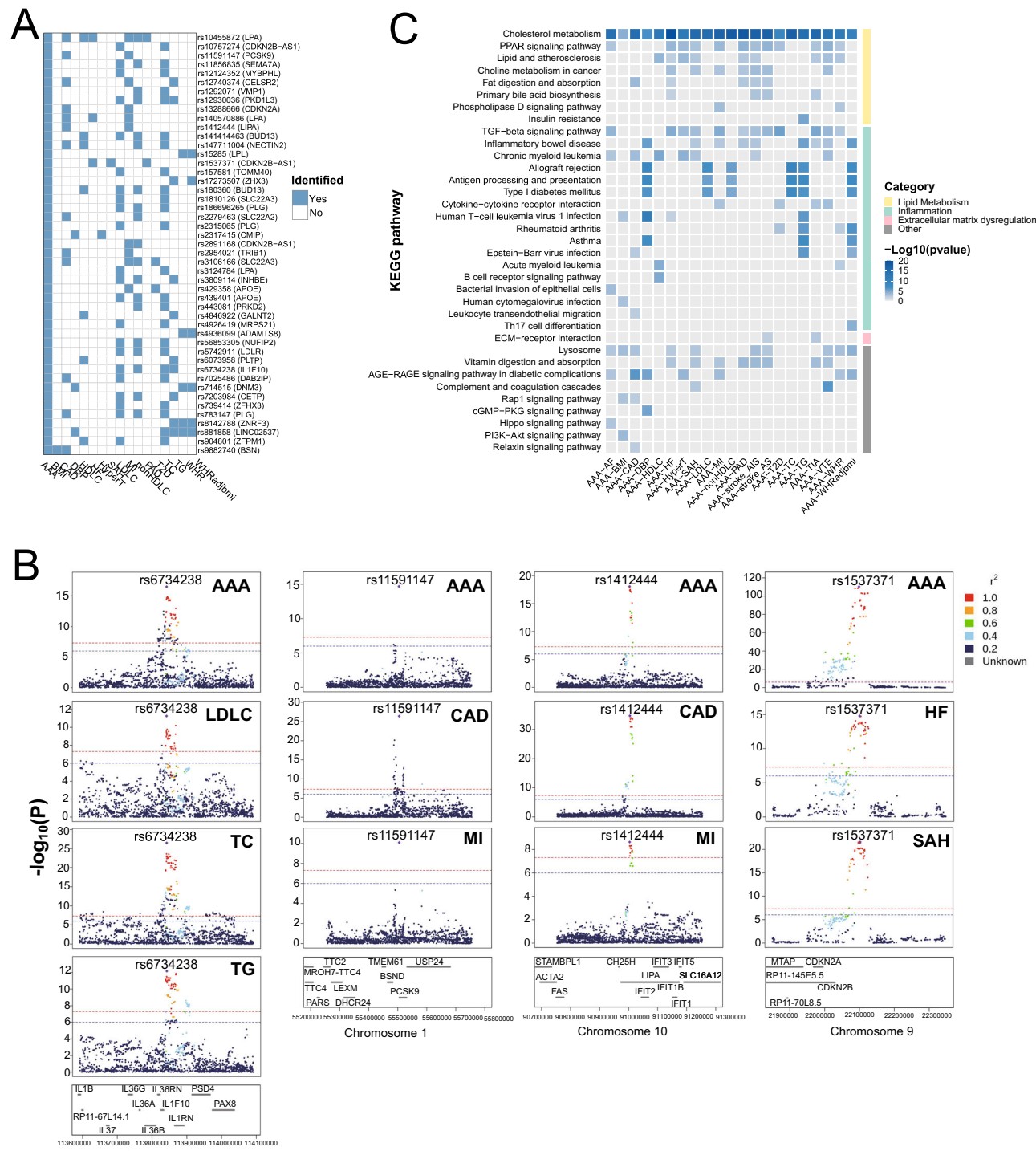

**Fig. 3 | The overall landscape of the pleiotropic associations across AAA and CMTs. A** 47 causal variants are shared by multiple traits, as identified by HyPrColoc. **B** LocusZoom plots of four causal variants for AAA and multiple other CMTs. These variants are also the lead SNVs in the interrogated regions. *P* values from original GWAS studies are presented. **C** KEGG pathway enrichment of the shared genes between AAA and CMTs, categorized by biological mechanisms. Only the top 15 enriched pathways passing hypergeometric test *P* < 0.05 in each trait pair were included. Source data are provided in this paper.

(Supplementary Fig. 7) were significantly enriched across many AAA-trait pairs, suggesting them as hubs for cardiac and metabolic functions (Fig. 5). Unique sharing is observed too. For example, muscle is only enriched by AAA and atrial fibrillation, the pituitary and brain are only enriched by AAA and BMI, and the pancreas is only enriched by AAA and HDL-C. While fibroblasts are broadly shared across traits, macrophages, and hepatocytes are more specific to AAA and lipid traits. Overall, these results align with the genes and

pathways, highlighting lipid metabolism and immunity over and again.

We additionally used SMR[40] and TWAS[39] to pinpoint gene-tissue effects for each CMT. Collectively, 116 genes were inferred for their directions of effect in tissues (Supplementary Fig. 8A-B). Here we highlight four most broadly shared genes: *CELSR2*, *PSRC1*, *LRP1*, and *NOC3L*. Both methods detected a negative relationship between CELSR2 expression in the liver with AAA and five other CMTs

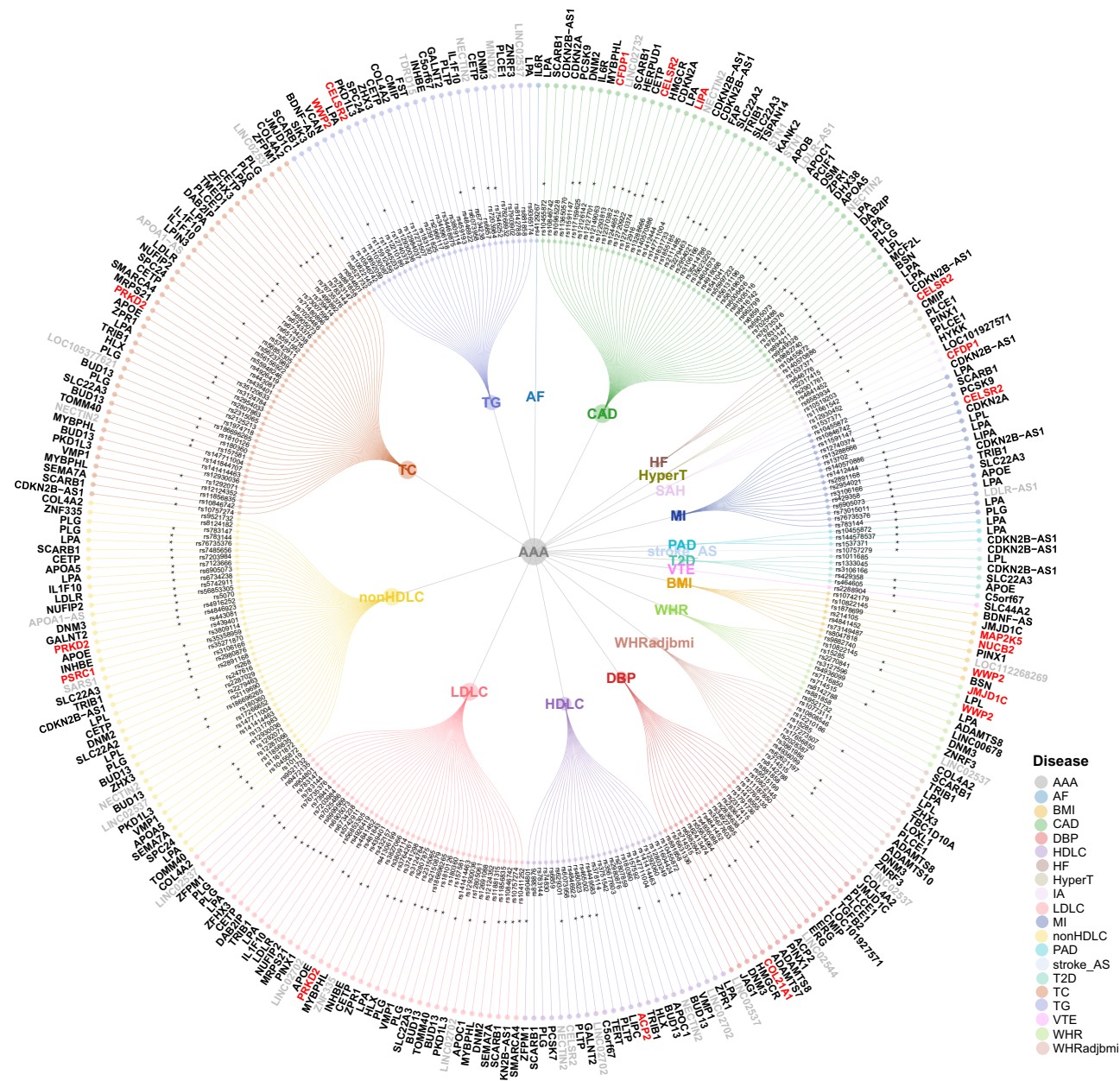

**Fig. 4 | Circular dendrograms presenting shared loci for AAA and CMTs.** The inner circle presents independent variants shared between AAA-trait pairs, with 177 shared causal variants marked in asterisks (posterior probability of H4 [PP.H4] > 0.7). The outer circle presents the genes inferred by Annovar for the shared variants. Genes are highlighted by colors to indicate overlap with the four gene identification methods: GCTA-fastBAT, MAGMA, TWAS, and SMR, with gray color for those not identified by any method, black color for those identified by at least one method, and red color for those identified by all four methods. Source data are provided in this paper.

(Supplementary Fig. 8C). Negative relationships were found for NOC3L expression in the skeletal muscle, and PSRC1 expression in the liver, whole blood, and esophagus mucosa, with AAA and numerous other CMTs. Meanwhile, LRP1 expression in the tibial artery was suggested for a positive relationship with AAA but a negative relationship with CAD.

**Drug for AAA with comorbid conditions**
Collectively we identified 405 disease genes shared by AAA and various CMTs. As cardiometabolic disorders often coexist, we used these genes to identify drugs for treating AAA with comorbidities. As such, we utilized a pathway paring score approach developed in our earlier study[47] to identify the best matching drugs and disease genes. Briefly,

we computed the pathological pathways for each trait pair based on their shared genes, and the pharmacological pathways for each candidate drug based on their affected genes recorded in large drug-gene databases, e.g., DrugCentral[48], DGIdb[49], and PharmGKB[50]. The candidate drugs were mainly derived from screening cardiovascular compounds that targeted any of the 405 disease genes. We also supplemented the list with those compounds used in clinical practice or clinical trials for treating AAA. Collectively, 33 candidate drugs distributed in 6 functional classes were examined, namely antihypertension (11 drugs), lipid-lowering (8 drugs), glucose-lowering (3 drugs), antiarrhythmics (1 drug), antithrombosis (4 drugs), and antioxidant (6 drugs). Most of these drugs have been approved to treat various cardiovascular diseases (Supplementary Data 5).

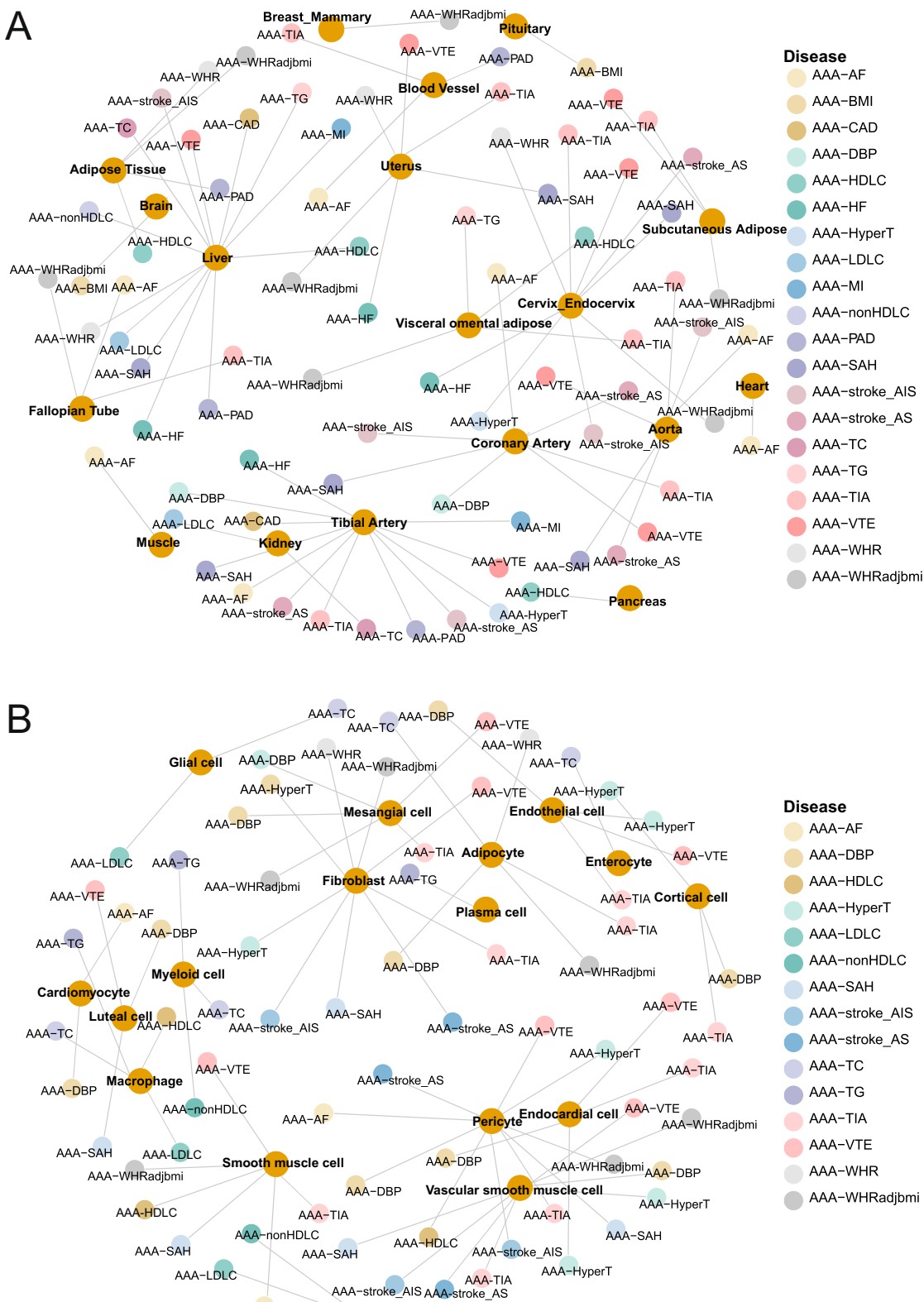

**Fig. 5 | Tissue and cell-type specificity inferred from the shared signals between AAA and CMTs. A** Enriched tissue types by the heritability or expression of the tissue-specific genes derived from GTEx. **B** Enriched cell types by the heritability of the cell type-specific enhancers derived from CATLAS, or expression of the cell type-specific genes in 11 single-cell transcriptome datasets. Source data are provided in this paper.

The best-matching drugs were defined with pairing scores >= 0.5 (Fig. 6). Close to half drugs, which were distributed in 4 functional categories, were suggested to treat AAA with hypertension. Therein amlopidine has the highest pairing score, followed by several antioxidants. Lipid-lowering drugs obtained high pairing scores for various trait pairs. Particularly, simvastatin and lovastatin both achieved high scores for AAA comorbid with CMDs, such as hypertension, MI, subarachnoid hemorrhage, transient ischemic attack,

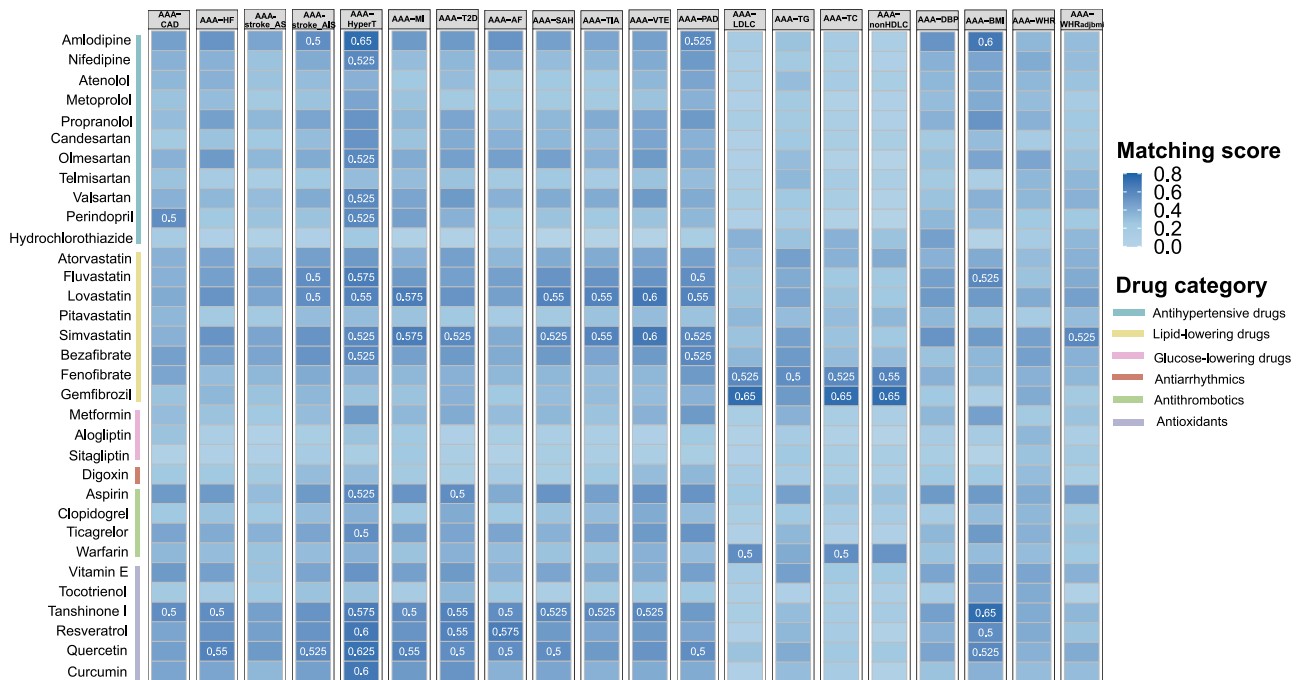

**Fig. 6 | Matching between disease pathological pathways, inferred from shared genes for each trait pair, and drug pharmacological pathways.** Matching scores greater than 0.5 are labeled.

venous thromboembolism, or peripheral artery disease. Interestingly, other lipid-lowering drugs are suggested for AAA with metabolic traits. For example, fenofibrate and gemfibrozil achieved high scores for AAA comorbid with LDL-C, nonHDL-C, triglycerides, or total cholesterol.

Notably, several herb-based antioxidants achieved high scores for various trait pairs too, including resveratrol, a stilbenoid polyphenol naturally enriched in red grapes; tanshinone I, a terpenoid exacted from the dry root of *Salvia miltiorrhiza* (Danshen); and quercetin, a flavonol found in many plants. Most of these herb products are in phase 3 clinical trials (Supplementary Data 5) and have shown potential in preventing and treating CVDs, including AAA[51–54]. Our analysis supports their extended application in treating comorbid conditions in AAA.

## Discussion

In this study, we discover extensive genetic associations between AAA and CMTs from GWAS summary statistics. Further analyses highlight the pleiotropic variants and genes, the biological pathways, and the types of cells and tissues that are shared by the trait pairs. All these findings help to elucidate the common genetic etiology between AAA and cardiometabolic disorders.

We discovered that among all CMDs outside of the aortic aneurysm family (i.e., AAA, TAA, and AA), CAD displays a consistently strong relationship with AAA. For example, it has the second highest genome-wide association ($r_g = 0.34$) and is the only trait with mutual causality with AAA ($OR_{AAA\rightarrow CAD} = 1.10$, $OR_{CAD\rightarrow AAA} = 1.23$). Epidemiological studies reported many risk factors common to AAA and CAD[55], and the two diseases tend to co-occur[56,57]. In our study, CAD shares the largest number of SNVs (N = 46), causal SNVs (N = 30), and disease genes (N = 50) with AAA. These shared signals were enriched for artery and liver tissues, reflecting their common malfunctions in artery and lipid metabolism. Artery-related diseases including peripheral artery disease ($r_g = 0.33$) and subarachnoid hemorrhage ($r_g = 0.32$), and cardiac-function-related diseases such as MI ($r_g = 0.38$) and heart failure ($r_g = 0.30$), also displayed top genome-wide associations with AAA, although no causal relationship was found, suggesting other risk factors may have confounded the associations.

Included in this study are the metabolic traits of lipids, adiposity, blood pressure, and glucose. By all levels of our inspection, lipid metabolism is most prominently shared. First, lipid traits rank as strong causal factors for AAA (OR = 1.46–1.73), next to hypertension, an established risk factor for AAA. Second, we observed clustering of the shared variants around lipid-related genes, including *LPA*, *CDKN2B-AS1* and others. Third, the most broadly shared genes between AAA and CMTs, i.e., *LRP1*, *PSRC1*, *CELSR2*, and *NOC3L*, are all lipid related and their up or down regulation was associated with CMTs (Supplementary Fig. 8). Fourth, cholesterol metabolism appeared as the most significantly enriched biological pathway. Fifth, liver, adipose tissue, hepatocytes, and adipocytes are most broadly and significantly enriched among the AAA-CMT trait pairs. These tissues and cell types are important players in lipid metabolism and regulation. Lastly, lipid-reducing drugs were strong candidates to treat AAA with various comorbid CMTs. These results reinforce the notion that predisposition to lipid malfunction is a strong feature in CMTs[58]. Our study also supports a higher burden of inherited dyslipidaemia in patients of AAA, and lowering LDL-C serves as a therapeutic strategy for preventing and managing AAA[59–62]. In comparison, glucose traits demonstrate neither genetic correlation nor causality to AAA. The relationship between glucose and AAA has been paradoxical. While epidemiological studies have reported an inverse correlation between the risk and growth rate of AAA and diabetic traits (e.g., HbA1c level[63], fasting glucose level[64], and T2D diagnosis[65]), no genetic correlation has been reported thus far[66], inviting further investigations. Lastly, among the blood pressure traits, only diastolic blood pressure displays a mild correlation ($r_g = 0.16$) and a weak causality ($OR_{DBP\rightarrow AAA} = 1.05$).

Several genes appeared repetitively in our analyses. *LPA* encodes lipoprotein(a), which is pro-atherosclerotic, pro-inflammatory, pro-thrombotic, and anti-fibrinolytic. Substantial evidence suggest that elevated lipoprotein(a) promotes CAD, MI, atherosclerosis, and aortic valve stenosis[67,68]. *CDK2B-AS1* encodes a long non-coding RNA that participates in inflammation as well as metabolism of lipids and carbohydrates, and has been linked to numerous CMDs and immune diseases[69,70]. *LRP1* encodes LDL receptor-related protein and plays

diverse roles in lipoprotein metabolism, endocytosis, cell growth, cell migration, inflammation, and apoptosis[44]. Furthermore, *CELSR2* and *PSRC1*, together with *SORT1*, form a *PRSC1-CELSR2-SORT1* axis which has been implicated in various CVDs[43,71]. *SORT1* encodes sortilin 1 that functions in lipid metabolism and immune responses, such as V-LDL secretion, LDL-C metabolism, PCSK9 secretion, inflammation, and formation of foam cells[72]. Finally, *NOC3L* is involved in adipocyte differentiation and glucose metabolism, and its decreased expression is associated with islet dysfunction[73].

We note that various disease genes in lipid metabolism are involved in immune responses too. Indeed, *LPA* is pro-inflammatory[74]; *CDK2B-AS1* is not only associated with numerous CMDs but also with immune diseases, such as idiopathic pulmonary fibrosis and inflammatory bowel disease[69,70,75]. Interestingly, statins, other than lowering lipids, are found to inhibit inflammation in AAA[76].

Indeed, there are abundant immune signals in our results. For example, IL-6 is an important cytokine in CVDs including AAA[77]. Enhanced IL-6 signaling will over-activate the JAK-STAT pathway, a critical pathway that affects many aspects of the mammalian immune system[78]. rs6734238 was reported to be associated with elevated circulating IL-6[79], whereas our analysis inferred this SNV as causal to AAA, LDL-C, total cholesterol, and triglycerides (Fig. 3B). We also identified two SNVs in the intronic regions of *IL6R*, rs4129267, and rs12126142, to be shared by AAA with atrial fibrillation and CAD, respectively. Furthermore, our pathway enrichment highlights the TGF-β signaling, which was shared by AAA and 12 CMTs (Fig. 3C). TGF-β regulates the differentiation and function of leukocytes and controls the type and scope of immune response[80]. Numerous studies have uncovered its importance in vascular smooth muscle cells (SMCs) and macrophages in the aneurysm development[81,82]. SMCs can transdifferentiate to foam cells, a crucial step in atherosclerosis[83]. In our analysis, both vascular SMCs and macrophages were enriched by several AAA-trait pairs. Indeed, various single-cell RNA-sequencing studies suggested them as essential cell types for AAA[84,85]. Our analysis reveals this close relationship also in genetic predisposition.

Overall, many of our results recapitulate the relationships of AAA with its risk factors and known disease markers, indicating our results captured the main components of AAA genetics. To confirm this, we analyzed the AAA single-trait GWAS loci. Many of the shared lipid-related genes are reproduced, and genes in various pathological mechanisms are connected (Supplementary Fig. 9–10). The most significantly enriched terms are lipid processes and cholesterol metabolism. The most enriched tissues are liver and blood vessels, and the most enriched cell types are fibroblasts, with a few others showing marginal enrichment, including endothelial cells, stromal cells, mesenchymal stem cells, macrophages, neutrophils, and monocytes. Therefore, the shared signals are the main signals in AAA genetics.

Finally, the shared disease genes are transformed into treatment proposals for treating AAA with comorbid conditions. Most drug candidates we discovered have been used to treat CVDs, and some are in clinical trials for repurposing to treat AAA, including atorvastatin, curcumin, metformin, and five other drugs (Supplementary Data 6). Note that numerous drugs with high matching scores in our analysis are not in clinical tests, inviting future studies for their therapeutic potential.

There are several limitations of this study. First, the GWAS data type enables analysis of common SNVs, but omits other variant types such as rare variants, the short insertions and deletions, and structural variants. Indeed, our previous whole-genome study identified a list of rare variants with strong predictability to AAA[18]. Second, we focus on genetics in this study as CMTs harbor high heritability in general, however, other factors such as epigenetics can also play important roles. As an example, smoking, as an established risk

factor of AAA, is known to cause vast epigenetic changes[86]. Third, CMTs cover a plethora of diseases and physiological traits, and those included in our analysis are only representative. Fourth, our inference of molecular and cellular mechanisms may be limited by the reference knowledgebases and databases. For example, we only deciphered the directions of effects for a quarter of the disease genes, due to the lack of variant-gene expression models in GTEx. With future improvements in the data types and references, we will gain further power to interpret results and infer the genetic mechanisms of AAA and other CMTs.

## Methods

### Study populations
We obtained summary statistics for AAA from a multi-ancestry meta-GWAS study (39,221 cases and 1,086,107 controls)[20]. Summary statistics for the other 32 CMTs were derived from UK Biobank (https://www.ukbiobank.ac.uk/), FinnGen (https://www.finngen.fi/en) or numerous large consortia[87–89]. Summary statistics for the counts and percentages of six immune cell traits were derived from the Blood Cell Consortium (BCX)[90], including white blood cell (WBC), basophil (BASO), eosinophil (EO), lymphocyte (LYMPH), monocyte (MONO), and neutrophil (NEUT), obtained from 563,085 participants of European ancestry. Information of these GWAS studies is provided in Supplementary Data 1.

### Genome-wide genetic correlation
We computed genome-wide genetic correlation between traits using linkage disequilibrium (LD) score regression (LDSC)[23]. Briefly, it quantifies the separate contributions of polygenic effects by examining the relationship between LD scores and test statistics of SNVs from GWAS summary results, producing a genetic correlation based on the deviation of chi-square statistics from the null hypothesis. LDSC also applies a self-estimated intercept during the analysis to account for shared subjects between studies. The derived estimates range from −1 to 1, with −1 indicating a perfect negative genetic correlation and 1 indicating a perfect positive genetic correlation. We used pre-computed LD scores obtained from -1.2 million common SNVs in the well-imputed HapMap3 European ancestry panel. A Bonferroni-corrected $P$ value threshold of 0.0015 (0.05/32) was used to define statistical significance.

### Genetic correlation by functional categories
We used LDSC to estimate genetic correlations between traits in 24 functional categories[91], e.g., transcribed regions, repressed regions, conserved regions, coding regions, promotors, enhancers, super-enhancers, introns, transcription factor binding sites (TFBS), DNaseI digital genomic footprinting (DGF) regions, DNase I hypersensitivity sites (DHSs), fetal DHS, untranslated regions (UTR), and histone marks (H3K4me1, H3K4me3, H3K9ac, and H3K27ac) from the Roadmap Epigenomics Project[91,92]. For each functional category, SNVs from the panel of HapMap3 European ancestry were assigned and LD scores were calculated, generating the "baseline model" (https://github.com/bulik/ldsc/wiki/Partitioned-Heritability). We downloaded them as the ldscore reference file to compute heritability enrichment and genetic correlation for the 24 functional categories.

### Mendelian randomization (MR) analysis
We used five MR methods to infer causal relationships between AAA and CMTs: inverse variance weighting (IVW)[93], MR-Egger[94], weighted median[95], MR-APSS[96], and CAUSE[97].

These methods utilize different assumptions about horizontal pleiotropy. Briefly, IVW assumes mean zero if uncorrelated pleiotropy is present, and such pleiotropy only adds noise to the regression of the meta-analyzed SNV effects with multiplicative random effects[93]. MR-Egger further allows for the presence of directional (i.e., non-zero

mean) uncorrelated pleiotropy and adds an intercept to the IVW regression to exclude such confounding effect[94]. The weighted median approach provides a robust estimate of causal effects even when up to 50% of genetic variants are invalid[95]. The recently published MR-APSS accounts for pleiotropy and sample structure, simultaneously[96]. Specifically, for decomposing the observed SNV effects, a foreground-background model is employed, in which the background model accounts for confounding factors (including correlated pleiotropy and sample structure) hidden in the GWAS summary statistics, and the foreground model performs causal inference while accounting for uncorrelated pleiotropy. CAUSE is a Bayesian MR method accounting for both correlated and uncorrelated pleiotropy[97]. Compared to the other MR methods, CAUSE further corrects correlated pleiotropy by evaluating the joint distribution of effect sizes from instrumental SNVs, assuming that the 'true' causal effect can influence all instrumental SNVs while correlated pleiotropy only influences a subset of them. CAUSE improves the power of MR analysis by including a larger number of LD-pruned SNVs with $P <= 1 \times 10^{-3}$ and provides a model comparison approach to distinguish causality from horizontal pleiotropy.

For selecting instruments, we used the genome-wide significance threshold $P = 5 \times 10^{-8}$ for IVW, MR Egger, and Weighted-median, the default threshold $P = 1 \times 10^{-3}$ for CAUSE, and the default threshold $P = 5 \times 10^{-5}$ for MR-APSS. We only selected independent SNVs (LD clumping $r^2 < 0.001$ within 1000 Kb using PLINK v1.9[98]) based on the European ancestry panel in the 1000 Genomes Project. In each LD block, we chose the variant with the smallest association $P$ value with the exposure. Further, we used PhenoScanner (http://www.phenoscanner.medschl.cam.ac.uk/) and GWAS Catalog (https://www.ebi.ac.uk/gwas/) to exclude SNVs associated with the outcome and its risk factors. IVW was used as the primary method, and the other four methods were used as sensitive analysis. A causal estimate was considered significant if it passed the $P$ value threshold in the primary analysis, i.e., IVW, and displayed a consistent direction of effect in all five MR methods. We used Cochran's Q-test to check for heterogeneity and MR-Egger intercept for horizontal pleiotropy. These MR analyses were performed in the R packages TwoSampleMR[99], MRAPSS[96] and CAUSE[97].

## Cross-trait meta-analysis

To identify pleiotropic loci shared between two traits, we performed a cross-trait meta-analysis of GWAS summary statistics using MTAG[24]. MTAG applies generalized inverse-variance-weighted meta-analysis for multiple traits; in addition, it accommodates potential sample overlap between GWAS. Its key assumption is that all SNVs share the same variance-covariance matrix of effect sizes among traits. As initially described[24], MTAG is a consistent estimator whose effect estimates have a lower genome-wide mean squared error than the corresponding single-trait GWAS estimates. In addition, association statistics from MTAG also yield stronger statistical power and little inflation of the FDR for each analyzed trait with high correlation[24].

As the assumptions in MTAG, i.e., equal SNV heritability for each trait and the same genetic covariance between traits, could be violated, we performed cross-phenotype association analysis (CPASSOC)[25] across traits as a sensitivity analysis. CPASSOC integrates GWAS summary statistics from multiple traits to detect shared variants while controlling population structure and cryptic relatedness[25]. It provides two test statistics, SHom and SHet. SHom is based on the fixed-effect meta-analysis and can be viewed as the maximum of weighted sum of trait-specific genetic effects. It is less powerful under the presence of between-study heterogeneity, which is common when meta-analyzing multiple traits. SHet is an extension to SHom with improved power that allows for heterogeneous effects of a trait from different study designs, environmental factors, or populations, as well as

heterogeneous effects for different phenotypes, which is more common in practice. SHet was thus adopted in our analysis. We applied PLINK clumping to obtain the independent SNVs (parameters: --clump-p1 $5 \times 10^{-8}$ --clump-p2 $1 \times 10^{-5}$ ---clump-r2 0.1 --clump-kb 1000). Significant pleiotropic SNVs were defined as variants with $P$ values in both GWAS studies and $P$ value in the meta-analysis (i.e., $P_{MTAG}$ & $P_{CPASSOC}$) $< 5 \times 10^{-8}$. We used ANNOVAR for functional annotation of the variants identified by MTAG and CPASSOC. The shared SNVs are visualized in a circular dendrogram using the R package ggraph.

## Fine-mapping credible set analysis

We identified a 99% credible set of causal variants by FM-summary (https://github.com/hailianghuang/FM-summary)[26], a Bayesian fine-mapping method. For each shared SNV identified in the cross-trait meta-analysis, we extracted variants within 500 Kb around the index SNV as input for FM-summary. FM-summary set a flat prior and produced a posterior inclusion probability (PIP) of a true association between a phenotype and a variant using the steepest descent approximation. A 99% credible set is equivalent to ranking the SNVs from largest to smallest PIPs and taking the cumulative sum of PIPs until it is at least 99%.

## Colocalization analysis

We used the R package coloc[27] to determine whether the association signals for AAA and CMTs co-localize. For each of the 203 shared SNVs between traits, we extracted the variants within 500 Kb of the index SNV and calculated the probability that the two traits share one common causal variant (H4). Loci with a probability greater than 0.7 were considered to colocalize. We estimated the posterior probability (PP) of multiple traits sharing the same SNV using a Bayesian divisive clustering algorithm implemented by HyPrColoc[28] (v.1.0.0) in R v.4.2.3.

## Gene-based association analysis

We used TWAS-fusion[39], SMR[40], MAGMA[41], and GCTA-fastBAT[42] to identify genes shared by AAA trait pairs. Input files for all four gene-level analyzes were the complete GWAS summary statistics from MTAG in the meta-analysis. In each method, the $P$ value threshold was adjusted by Bonferroni correction.

TWAS identifies tissue-specific gene-trait associations by integrating GWAS with cis-SNVs based gene expression model[39,100]. We conducted TWAS using the FUSION software[39] based on 43 postmortem tissue expression profiles in GTEx (version 6, with pre-computed models)[45].

Summary-data-based Mendelian Randomization (SMR) analysis integrates GWAS and eQTL studies to identify genes whose expression levels are associated with a complex trait due to pleiotropy or causality[40]. A significant SMR association could be explained by a causal effect (i.e., the causal variant influences disease risk via changes in gene expression), pleiotropy (i.e., the causal variant has pleiotropic effects on gene expression and disease risk), or linkage (i.e., different causal variants exist for gene expression and disease). SMR implements the HEIDI-outlier test to distinguish pleiotropy from linkage. We implemented SMR using cis-eQTL summary data for whole blood from eQTLGen[101], a meta-analysis of 31,684 blood samples, and from GTEx V8 for 9 relevant tissues, including artery aorta, adipose subcutaneous, artery coronary, artery tibial, heart atrial appendage, heart left ventricle, kidney cortex, liver, and whole blood. Genes associated with AAA-trait pairs were defined as $P_{SMR}$ passing the Bonferroni-corrected thresholds and $P_{HEIDI} > 0.05$.

MAGMA[41] (Multi-marker Analysis of GenoMic Annotation) is a fast and flexible method that uses a multiple regression approach to properly incorporate LD between markers and detect multi-marker effects. We ran MAGMA with default parameters, with the European ancestry panel in the 1000 Genomes Project (Phase 3) as the LD reference.

We applied a fourth approach, GCTA-fastBAT[42], a fast set-based association analysis. In brief, it calculates the association *P* value for a set of SNVs from an approximated distribution of the sum of $\chi^2$-statistics over all SNVs using GWAS summary data and LD correlations from a reference sample set with individual-level genotypes[42]. We used the European ancestry panel in the 1000 Genomes Project (Phase 3) as the LD reference.

### Stratified LD score regression for tissue and cell type specificity

We used LD score regression applied to specifically expressed genes (LDSC-SEG)[102] for tissues or cell types to test for heritability enrichment. For tissues, pre-computed LD scores from GTEx[103], which contained gene expression data for 53 tissues, were provided by LDSC-SEG and used in our analysis. We also obtained the activity profile of candidate cis-regulatory elements (cCREs) in 222 cell types from CATLAS[46]. We mapped the genotypes of European ancestry in the 1000 Genomes Project to the cell type-specific cCREs and calculated the cell type-specific ldscore. We applied FDR correction for each dataset respectively to account for multiple testing and considered FDR corrected *P* < 0.05 as significant.

### GTEx Tissue Specific Expression Analysis (TSEA)

We conducted tissue-specific expression analysis (TSEA)[104] on the shared genes against the RNA-seq data in GTEx, which contained gene expression profiles of 1839 samples from 45 different tissues derived from 189 post-mortem subjects. We merged the shared genes for each trait pair from the four gene-based analyses to derive a collection of shared genes. Hypergeometric tests are used to determine if tissue-specific genes are enriched in the input genes. We used Benjamini–Hochberg correction to account for multiple testing (FDR < 0.05).

### Cell-type-specific enrichment analysis (CSEA)

We performed cell-type specific enrichment (CSEA) on the shared genes using WebCSEA[105]. WebCSEA provides a gene set query against tissue-cell-type (TCs) expression signatures of 11 single-cell gene expression datasets[106–112]. Specifically, Dai et al.[105] collected more than 5.5 million cells from 111 tissues and 1355 TCs, filtered out the low expression genes, and used an in-house t-statistic-based method "deTS" to train the tissue-cell-type signature genes. Genes with the top 5% t-statistic scores in focal cell type are defined as cell type-specific genes. We conducted Fisher's exact test to assess whether the shared genes for each trait pair are overrepresented with the cell type-specific genes.

### Over-representation enrichment analysis

We used the "clusterProfiler" package[113] to perform GO (Gene Ontology) and KEGG (Kyoto Encyclopedia of Genes and Genomes) pathway enrichment analysis on the genes. Benjamini–Hochberg procedure was used to account for multiple tests (FDR < 0.05).

### Drug target analysis

We defined the genes identified by all four gene-based analyses as the disease genes for each AAA-trait pair and thus obtained 405 genes collectively. For deriving drugs that match the best with the genes, we leveraged the biological pathway analysis. First, we applied cluster-Profiler to the shared genes to compute the pathological pathways enriched for each AAA trait pair. Next, we carried multiple steps to derive candidate drugs for scrutinization: (1) queried three large drug-gene databases, DrugCentral[48], DGIdb[49], and PharmGKB[50], for drugs that target any of the 405 candidate genes. This initial screening led to about 1200 compounds, most of which were initially designed for treating cancer; (2) limited the compounds to those already in use by standard clinical practices for treating cardiovascular diseases, which drastically shrank the list to 21 drugs; and (3) supplemented the list

with 12 drugs used in clinics or proposed by clinical trials for treating AAA, such as amlodipoine, pitavastatin, and vitamin E. Collectively, 33 candidate drugs were derived. Then, for each drug, we queried the three drug-gene databases again for all their affected genes and computed their enriched pharmacological pathways in clusterProfiler. Finally, we calculated the pairing scores between the pathological pathways of the cross-trait and the pharmacological pathways of the drug[47].

### Inquiry of clinical trial information

We queried the ChEMBL database (https://www.ebi.ac.uk/chembl) for information on action targets, action types, indications, and clinical development activities associated with the 33 candidate drugs. We also searched the https://www.ClinicalTrials.gov website for additional information on clinical trials of drugs.

### Software packages used

Publicly available software was used to perform the analyses. Analysis software includes LDSC, PLINK, LAVA, TwoSampleMR (https://mrcieu.github.io/TwoSampleMR), CAUSE, MR-APSS, MR-BMA (https://github.com/verena-zuber/demo_AMD), MTAG: (https://github.com/JonJala/mtag), CPASSOC (http://hal.case.edu/~xxz10/zhu-web), Coloc (https://github.com/chr1swallace/coloc), HyPrCo-loc (https://github.com/cnfoley/hyprcoloc), FM-summary (https://github.com/hailianghuang/FM-summary), FUSION, MAGMA, SMR (https://cnsgenomics.com/software/smr/#Overview), GCTA-fastBAT (https://yanglab.westlake.edu.cn/software/gcta/#fastBAT), TSEA, FUMA, and WebCSEA (https://bioinfo.uth.edu/webcsea).

### Reporting summary

Further information on research design is available in the Nature Portfolio Reporting Summary linked to this article.

## Data availability

GWAS summary statistics for this study were listed in Supplementary Data 1. Whole blood-based *cis*-eQTL summary statistics (the dataset "Full cis-eQTL summary statistics") were downloaded from eQTLGen (https://molgenis26.gcc.rug.nl/downloads/eqtlgen/cis-eqtl/2019-12-11-cis-eQTLsFDR-ProbeLevel-CohortInfoRemoved-BonferroniAdded.txt.gz). eQTL summary data of multiple tissues (the dataset "GTEx_Analysis_v8_eQTL.tar") were downloaded from GTEx (https://gtexportal.org/home/downloads/adult-gtex/qtl), specifically, https://storage.googleapis.com/adult-gtex/bulk-qtl/v8/single-tissue-cis-qtl/GTEx_Analysis_v8_eQTL.tar). Source data are provided with this paper.

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

## Acknowledgements

S.Z. is a Ph.D. student at School of Life Sciences, Fudan University. This study was supported by the research grants awarded to C.P. from National Natural Science Foundation of China (No. 32270626) and Greater Bay Area Institute of Precision Medicine (Guangzhou) (I0005 and R2001). We thank members of the Laboratory of Intelligent Computing in Biomedicine in the Greater Bay Area Institute of Precision Medicine (Guangzhou) for insightful discussions and suggestions.

## Author contributions

C.P., S.Z., and P.S.T. collectively designed the study. S.Z. and C.P. performed bioinformatic and statistical analyses and generated the figures and tables. P.S.T. and C.P. provided critical biological insight in interpreting the results. C.P. and S.Z. drafted the manuscript. All authors critically reviewed the manuscript.

## Competing interests

The authors declare no competing interests.
