## [Peer review file · Nature Communications]

Abdominal Aortic Aneurysm and Cardiometabolic Traits Share Strong Genetic Susceptibility to Lipid Metabolism and InflammationREVIEWER COMMENTS

Reviewer #1 (Remarks to the Author):

In their manuscript, Zheng et al. analyze the potential shared genetic susceptibility between abdominal aortic aneurysm and a extensive list of cardiometabolic traits. They describe the genetic correlation and perform bidirectional Mendelian Randomization with the aim to infer causality between these traits. Moreover, they define shared variants and potential causal genes through colocalization, and show the enrichments for biological pathways and cell and tissue types for the analyzed trait pairs. Finally, they conduct an in sillico screen of drug candidates for the selected genes shared by AAA and cardiometabolic traits.

The data, methodology and analytical approach are sound. The description of the results is clear and accordingly to the presented data. The highest correlation and biggest significance in the MR analysis occurs with other vascular diseases and lipid and cholesterol related traits. The shared genes, pathways and tissues support the notion of the relevance of lipid and cholesterol traits on AAA risk. Concomitantly, cholesterol lowering drugs like statins are found among the most relevant compounds described in the drug screen.

While the premise of the study is interesting and the work is well done, the results are not utterly novel. As the authors themselves state (starting page 12, line 331): "Overall, many of our results recapitulate the relationship of AAA with its risk factors and known disease markers, ...". In fact, one of the most sticking results is the lack of association or causality of AAA with glucose traits like FG, FI, T2D, specially given their association with CAD, and cholesterol and lipid metabolism deregulation. It would be interesting to know the thoughts of the authors on why this is the case in their results. In addition, the well documented connection between high cholesterol and lipid levels and the risk of developing CAD (statins are one of the main drugs prescribed to treat CAD- Are they prescribed to prevent AAA?) is thought to occur through the effect of this metabolites on the vessel wall, and not so much through common variants that affect cholesterol levels and vessel wall biology independently at the same time. Would be interesting to know if the authors have a sense if this could also be the case in their study. Making the question in a different way: what part

of the found commonalities between traits is acting simultaneously regulating the metabolism AND the vessel wall biology? Are the found common genes mainly causal because of their effect on cholesterol and lipid levels?

Minor comments:

1) I will suggest giving the title of the manuscript a more focused approach describing in general terms what is actually the shared traits with AAA. Currently, in my opinion, the title leads to confusion as it seems that the overlap between CMT and AAA is larger than the results presented in the manuscript.

2) Page 4, line 110 “GWAS provided comprehensive mutation profiles for dozens of cardiometabolic traits (CMTs)”. Please substitute the word mutation for either variant or polymorphism.

Reviewer #2 (Remarks to the Author):

Zheng et al use summary statistics from a recent GWAS on AAA to identify causal variants, genes and pathways for AAA. Genetic correlation analysis identify cardiometabolic traits with shared genetic factors. In combination with GTEx data, the authors aim to identify tissue specific effects driving the disease and drug targets.

The manuscript is well written and the work of interest of those working on cardiometabolic diseases but also for those interesting on pipelines of statistical genetic methods to understand complex traits. There are a few things that not clear from the methods and the interpretation of the result, so I hope the authors would clarify the following points:

1) The main objective of the work is not clearly stated in the manuscript. As it is right now they highlight as objective to identify drug targets for AAA. But the work seems to focus on finding candidate causal genes, which of course have the potential to be therapeutic targets, but the reasoning around why their strategy is good/novel/useful is not clear. I

believe the authors should revise the manuscript to make whatever point they want to focus on more clearly.

For example, they wrote: “We aim to construct a map of comprehensive relationship, as well as to provide details such as shared SNVs, genes, and pathways in the cell type and tissue context. Importantly, this comorbidity landscape offers valuable information for prioritizing drugs that target shared genes.”

Why or how can the comorbidity landscape help prioritizing drugs? Why do you need a “map”? Why from those other traits?

2) The authors perform an overwhelming number of analyses, often with multiple different methods to then proceed to describe the consensus results of those different methods and any difference they found or a selection of each. So either the authors believe only the findings that agree across different methods, or all the results independently form the analysis that produce them. This lack of consistency is confusing and it seems as if the authors chose results as it is convenient and based on some unknown criteria.

Detailed examples:

2.1) Paragraph starting on line 174 describe genes annotated to SNVs based on proximity, an arbitrary criteria that ignores pleiotropy. Right after the authors proceed to describe four additional approaches to actually identify and test candidate genes mediating the activity of the candidate causal SNVs. The genes here described are not supported by any analysis, but they compared them in the next paragraph and with the next results as if this was 5th type of analysis to identify genes that once agreed with the other four means those genes are “more valid”.

So five approaches (four methods + proximity) are used to identify candidate genes and consensus is used here as a sign of validation at first, just to proceed to talk about all results (line 195) because consensus is no longer necessary.

Why do the authors assume consensus means the results are more believable? The TWAS models from FUSION used GTExv6. I presume the authors used pre-computed models and these were on v6 (all 43 tissues), if so please state this in the methods. Otherwise, explain why for SMR they chose GTEx v8 and limit the analyses to 9 tissues (?). Finally, the MAGMA criteria for identifying genes is mostly based on proximity and bias to more studies and better annotated genes. So there is a lot of room for these methods to find different genes, and it is nearly impossible for the authors or the reader to know which one may be more informative. Criteria to talk about some genes and no other need to be clearer.

2.2) They used PLINK clumping to find independent SNVs for the functional annotation, then FM-summary to find a credible set for fine mapping. Why two methods to produce the same type of output?

3) Please explain the following sentence (line 143): “Repressors, enhancers and promoters tend to have stronger than the genome-wide correlations (Supplementary Fig. 1), suggesting transcription regulation is genetically shared.”. The authors assumed the reader is fully familiar with LDSC and that “stronger than the GW correlations” has a full meaning. It does not. It is not clear how the authors reach the conclusion that transcription regulation is shared (across traits?).

4) Lines 143-145: “Among the 47 shared causal variants, only four were local lead SNVs (Fig. 3B), i.e., having the smallest GWAS P values, whereas the rest were located near the lead SNVs, suggesting the importance of fine-mapping.”. What are the authors trying to tell us here? Fine mapping methods were developed to identify causal variants resolving issues such as LD contamination, it is not the result of their analyses.

5) Line 281: “[...] hypertension, which is a disease rather than a physiological trait.”. This sentence makes no sense. Diseases and traits are all the results of genetic and environmental factors, so I am not sure why the authors want to make a distinction here that have no added value.

6) The authors are assuming some kind of directionality in some of their results. For example, line 291: “These results reinforce the notion that predisposition to lipid

malfunction is a strong feature in CMTs” The results do not indicate directionality. There is no information that would be derived about whether lipids are malfunctioning (?) or not in relation to CMTs. If there is some form of directionality, it needs to be made clear.

Reviewer #3 (Remarks to the Author):

What are the noteworthy results?

This review is a comprehensive overview of the genetic findings that overlap between cardiovascular diseases and aneurysm disease. The findings do of course by the nature of being a review correspond well with the reports in the field, and maybe do not share great new findings, but on the other hand clarify and enlighten important shared areas within genetics for these patientgroups.

Will the work be of significance to the field and related fields?

This review is comprehensive and potentially can be important and contribute to future lines to follow especially in the AAA area.

Does the work support the conclusions and claims, or is additional evidence needed?

It could be a more broad inclusion of more relevant clinical papers supporting for example the influence by lipidpathology or statin therapy as support of the findings. There is a general lack of clinical aspect. An example could be Line 267-268; many clinical papers also support this. Please include and thereby the clinical relevance of the findings will be upgraded!

Are there any flaws in the data analysis, interpretation and conclusions?

Do these prohibit publication or require revision? no apparent for me.

Is the methodology sound? Does the work meet the expected standards in your field?

Yes, this appears well performed, and cover and include all the more recent published relevant work in the field.

There is possibly a lack of epigenetic reflections; we know as clinicians the profound influence by smoking on disease development and expansion for aneurysms, could this be attributed some comments/other analysis?

Is there enough detail provided in the methods for the work to be reproduced?

Specific review comments:

The text which refers to more clinical data or are disease-related are sometimes not correct or clear.

Abstract: "AAA presents abnormal metabolism – would never be used in a clinical context reporting on AAA, what do the authors mean?"

Lack SNV in abbreviation list.

Line 81-82 The authors could benefit from more updating to more contemporary references for AAA disease. The prevalence rate is rarely 3-9%, even if it refers to a recent review in EHJ, there are many more contemporary prevalence reports rather going down to 0.5 (in women) to 4% in men. Please revise and review.

Line 84. Question this report, there is much stronger evidence for rupture rate in other papers. Please revise. (3)

Line 84-85 This paper (4) is also a bit odd as it refers to a review but reports it as one number please revise and maybe add more international and clear report-data?

Line 86: Pathologically is this really a good choice of word here? The pathogenesis is...?

Section line 86-100 Please look into your choice of reporting the literature. The reader gets the impression that you have a certified causality between hyperlipidemia and AAA. It is also strange to read that line 92-95 that one has hypercholesterolemia in AAA patients, as if this a certified association in all patients, which it is not. Please revise.

Line 272 "... consistent with their common features of artery malfunction and atherosclerosis" really unclear of the meaning of this?

Other/more traits should then be included? Please revise and explain?

rad 289-290 Unsure of the meaning of this kind of text: " were suggested to treat many AAA with comorbid disease", please revise and change.

rad 291 "In comparison, glucose traits demonstrate neither correlation of causality to AAA" -- perhaps some discussion about this in comparison to the literature?

Last section; line 555-567

Very interesting and relevant analysis. However there is a lack of reflection on these drugs as compared to already performed or planned RCT trials in the field. Please add the corresponding trials if there are some in this list? Such as statins etc and correlate to your suggestions?

Thank you for inviting me to review this manuscript.

Response to reviewers

REVIEWER #1

In their manuscript, Zheng et al. analyze the potential shared genetic susceptibility between abdominal aortic aneurysm and a extensive list of cardiometabolic traits. They describe the genetic correlation and perform bidirectional Mendelian Randomization with the aim to infer causality between these traits. Moreover, they define shared variants and potential causal genes through colocalization, and show the enrichments for biological pathways and cell and tissue types for the analyzed trait pairs. Finally, they conduct an *in silico* screen of drug candidates for the selected genes shared by AAA and cardiometabolic traits.

The data, methodology and analytical approach are sound. The description of the results is clear and accordingly to the presented data. The highest correlation and biggest significance in the MR analysis occurs with other vascular diseases and lipid and cholesterol related traits. The shared genes, pathways and tissues support the notion of the relevance of lipid and cholesterol traits on AAA risk. Concomitantly, cholesterol lowering drugs like statins are found among the most relevant compounds described in the drug screen.

While the premise of the study is interesting and the work is well done, the results are not utterly novel. As the authors themselves state (starting page 12, line 331): "Overall, many of our results recapitulate the relationship of AAA with its risk factors and known disease markers, ...". In fact, one of the most sticking results is the lack of association or causality of AAA with glucose traits like FG, FI, T2D, specially given their association with CAD, and cholesterol and lipid metabolism deregulation. It would be interesting to know the thoughts of the authors on why this is the case in their results. In addition, the well documented connection between high cholesterol and lipid levels and the risk of developing CAD (statins are one of the main drugs prescribed to treat CAD- Are they prescribed to prevent AAA?) is thought to occur through the effect of this metabolites on the vessel wall, and not so much through common variants that affect cholesterol levels and vessel wall biology independently at the same time. Would be interesting to know if the authors have a sense if this could also be the case in their study. Making the question in a different way: what part of the found commonalities between traits is acting simultaneously regulating the metabolism AND the vessel wall biology? Are the found common genes mainly causal because of their effect on cholesterol and lipid levels?

Response:

We thank the reviewer for the concise and accurate summary of our study. The reviewer asks about two very important risk factors for atherosclerotic CVD (ASCVD) in general: glucose and lipids, which we address in the following.

1) Lack of association or causality of AAA with glucose traits including T2D.

This is an interesting and puzzling finding that aroused common interest, as another reviewer asked about it too. Several other genetics-based studies reported similar results,

i.e., no genetic correlation was found between AAA and T2D (ref 1,2). On the contrary, epidemiological studies observed a positive association between T2D and CAD, and an inverse association between T2D and AAA formation/expansion (ref 3,4). That is, even though AAA and CAD both belong to atherosclerotic CVD and are closely related genetically, they display opposite directions of co-occurrence with T2D, indicating a complex triangular relationship among AAA, CAD and T2D. How T2D protects against AAA remains unknown but suggested mechanisms include glycation on ECM remodeling, immune response and intraluminal thrombus formation, among others, may play an important role (ref 4,5). Clearly further basic investigation is needed to understand it thoroughly. In our revised manuscript, we add related comments in the Discussion section:

Page 10, lines 276-281: “In comparison, glucose traits demonstrate neither genetic correlation nor causality to AAA. The relationship between glucose and AAA have been paradoxical. While epidemiological studies have reported an inverse correlation between the risk and growth rate of AAA and diabetic traits (e.g., HbA1c level ⁶³, fasting glucose level ⁶⁴, and T2D diagnosis ⁶⁵), no genetic correlation has been reported thus far ⁶⁵, inviting further investigations.”

REFERENCE:

1. Morris DR, et al. Genetic Predisposition to Diabetes and Abdominal Aortic Aneurysm: A Two Stage Mendelian Randomisation Study. *Eur J Vasc Endovasc Surg.* 2022 Mar;63(3):512-519.
2. van 't Hof FN, et al. Genetic variants associated with type 2 diabetes and adiposity and risk of intracranial and abdominal aortic aneurysms. *Eur J Hum Genet.* 2017 Jun;25(6):758-762.
3. De Rango P, et al. Diabetes and abdominal aortic aneurysms. *Eur J Vasc Endovasc Surg.* 2014 Mar;47(3):243-61.
4. Raffort J, et al. Diabetes and aortic aneurysm: current state of the art. *Cardiovasc Res.* 2018 Nov 1;114(13):1702-1713.
5. Morris DR, et al. Genetic Predisposition to Diabetes and Abdominal Aortic Aneurysm: A Two Stage Mendelian Randomisation Study. *Eur J Vasc Endovasc Surg.* 2022 Mar;63(3):512-519.

2) Common causal genes that impact lipids and vessel wall biology.

Both lipids and vessel wall biology are the focus of AAA studies. Vascular smooth muscle cells and endothelial cells are the major cell components of the aorta, and AAA is manifested with vessel inflammation and loss of aortic elasticity. Separately, lipids are considered a significant risk factor for CMD. While there is no therapy for AAA, statins are usually prescribed to AAA patients for secondary prevention (ref 6).

In our AAA single trait analysis (**Supplementary Fig. 10**), we observed strong signals of lipids and vessel wall biology, both in pathway enrichment (lipids: lipoprotein/cholesterol metabolism; vessel wall biology: TGF-beta signaling/ECM-receptor interaction) and tissue/cell type specificity (lipids: liver, adipocytes; vessel wall biology: artery, blood vessel, vascular smooth muscle cells, fibroblasts, and so on). Interestingly, for the shared genetic signals between AAA and other cardiometabolic traits, which is a subset of the AAA genetics, we observed a very strong signal in lipids, and less so in vessel wall biology (**Fig.3b, Supplementary Fig. 5 and Supplementary Fig. 9**). To begin to understand the simultaneous regulation of lipid metabolism and the vessel wall biology, we did the following:

The AAAgen GWAS study published last year (ref 7), which is the source of our AAA GWAS data, had carefully curated 45 GWAS genes related to vessel wall biology. We overlap our cross-trait disease genes involved in cholesterol metabolism (n=38) with

these vessel wall genes, and identified 9 genes in total that are common for both cholesterol metabolism and vessel wall biology. Distribution of the 38 cholesterol metabolism genes is displayed below. Through this we estimate that roughly 1/4 cholesterol genes are involved in vessel wall biology too.

REFERENCE:

- Wemmelund H, et al. Statin use and rupture of abdominal aortic aneurysm. *Br J Surg.* 2014 Jul;101(8): 966-975.
- Roychowdhury, T., Genome-wide association meta-analysis identifies risk loci for abdominal aortic aneurysm and highlights PCSK9 as a therapeutic target. *Nat Genet.* 2023 Nov;55(11):1831-1842.

Minor comments:

1) I will suggest giving the title of the manuscript a more focused approach describing in general terms what is actually the shared traits with AAA. Currently, in my opinion, the

title leads to confusion as it seems that the overlap between CMT and AAA is larger than the results presented in the manuscript.

Response:

Thank you for your suggestion! We change the title to “**Abdominal Aortic Aneurysm and Cardiometabolic Traits Share Genetic Susceptibility to Lipid Metabolism and Inflammation**”.

2) Page 4, line 110 “GWAS provided comprehensive mutation profiles for dozens of cardiometabolic traits (CMTs)”. Please substitute the word mutation for either variant or polymorphism.

Response:

Indeed, the word *variant* would be a more precise description. We changed it accordingly in our revised manuscript:

Page 4, lines 91-93: “Similarly, recent GWAS provided comprehensive variant profiles for dozens of cardiometabolic traits (CMTs), bringing the disease understanding to a new level.”

REVIEWER #2

Zheng et al use summary statistics from a recent GWAS on AAA to identify causal variants, genes and pathways for AAA. Genetic correlation analysis identify cardiometabolic traits with shared genetic factors. In combination with GTEx data, the authors aim to identify tissue specific effects driving the disease and drug targets.

The manuscript is well written and the work of interest of those working on cardiometabolic diseases but also for those interesting on pipelines of statistical genetic methods to understand complex traits. There are a few things that not clear from the methods and the interpretation of the result, so I hope the authors would clarify the following points:

1) The main objective of the work is not clearly stated in the manuscript. As it is right now they highlight as objective to identify drug targets for AAA. But the work seems to focus on finding candidate causal genes, which of course have the potential to be therapeutic targets, but the reasoning around why their strategy is good/novel/useful is not clear. I believe the authors should revise the manuscript to make whatever point they want to focus on more clearly.

For example, they wrote: “We aim to construct a map of comprehensive relationship, as well as to provide details such as shared SNVs, genes, and pathways in the cell type and tissue context. Importantly, this comorbidity landscape offers valuable information for prioritizing drugs that target shared genes.”

Why or how can the comorbidity landscape help prioritizing drugs? Why do you need a “map”? Why from those other traits?

Response:

We thank the reviewer for carefully reviewing our methods and interpretation of results. These constructive comments prompted us to revise the manuscript accordingly, which we explain by answering to the reviewer's comments point by point.

The main objective of our study is to identify shared genetic signals between AAA and a plethora of cardiometabolic traits. The foundation lays in the high heritability of CMTs, for example, 70% for AAA and 60% for CAD, suggesting genetics as a powerful tool to decode the biology of cardiometabolic diseases. The motivation to compare AAA with other CMTs came from the observation of high co-occurrence of CVDs. We hypothesized that placing AAA in context ("a map") with other cardiometabolic traits, especially the main categories of metabolism including glucose, lipids, adiposity and blood pressure, would help us understand underlying biology and enlighten the discovery of therapeutic targets. Indeed, the map shows that cholesterol metabolism as the most enriched signal shared by AAA and other CMTs, and anti-lipid agents are suggested as broadly beneficial for various AAA trait pairs. That said, however, we'd like to stress that the drugs reported in our studies are identified by analyzing shared disease genes for co-occurring diseases. The best application scenario is for treating the comorbid diseases, i.e., AAA plus the other disease under the consideration. We did not mean to identify drugs for treating AAA alone. To derive such candidate drugs, we utilized a network pharmacology algorithm that considers matching between the biological pathways of the diseases and biological pathways impacted by the drugs. Now we have revised the manuscript to better explain the objective of our study:

Pages 4-5, lines 94-98: "In this study, we leverage these large GWAS data to identify genetic factors shared by AAA and CMTs. We aim to identify shared SNVs and genes, as well as the enriched pathways, cell types, and tissues. Importantly, these results offer valuable information for prioritizing drugs that target shared genes for treating AAA with comorbid conditions."

2) The authors perform an overwhelming number of analyses, often with multiple different methods to then proceed to describe the consensus results of those different methods and any difference they found or a selection of each. So either the authors believe only the findings that agree across different methods, or all the results independently form the analysis that produce them. This lack of consistency is confusing and it seems as if the authors chose results as it is convenient and based on some unknown criteria.

Detailed examples:

2.1) Paragraph starting on line 174 describe genes annotated to SNVs based on proximity, an arbitrary criteria that ignores pleiotropy. Right after the authors proceed to describe four additional approaches to actually identify and test candidate genes mediating the activity of the candidate causal SNVs. The genes here described are not supported by any analysis, but they compared them in the next paragraph and with the next results as if this was 5th type of analysis to identify genes that once agreed with the other four means those genes are "more valid".

So five approaches (four methods + proximity) are used to identify candidate genes and

consensus is used here as a sign of validation at first, just to proceed to talk about all results (line 195) because consensus is no longer necessary.

Why do the authors assume consensus means the results are more believable? The TWAS models from FUSION used GTEx v6. I presume the authors used pre-computed models and these were on v6 (all 43 tissues), if so please state this in the methods. Otherwise, explain why for SMR they chose GTEx v8 and limit the analyses to 9 tissues (?). Finally, the MAGMA criteria for identifying genes is mostly based on proximity and bias to more studies and better annotated genes. So there is a lot of room for these methods to find different genes, and it is nearly impossible for the authors or the reader to know which one may be more informative. Criteria to talk about some genes and no other need to be clearer.

Response:

Thank you for pointing out the lack of clarity on gene selection criteria. Assigning GWAS SNPs to genes has been a challenging task, and recent studies tended to integrate multiple methods for this purpose (ref 1, 2, 3). In our study, we first applied the closest distance approach in the section “Cross-trait loci and causal variants” for highlighting the clustering of SNPs close to or on lipid genes (page 6-7, lines 152 – 160). Next for defining GWAS genes (page 7, lines 162 – 172), we utilized a comprehensive approach of combining four methods, essentially from two features: proximity + sum- χ^2 method (GCTA-fastBAT and MAGMA) and eQTLs (TWAS-Fusion and SMR). The proximity approach ignored pleiotropic effects, as the reviewer pointed out, so it is expected to have higher false positive rate; whereas the four-method approach required additional features, it is likely to reduce false positives, although it may be too conservative to require eQTL effects and hence missing GWAS genes that do not function via regulating gene expression. Weighing the *pros* and *cons*, we decided to go with the conservative four-method strategy for defining disease genes. For pathway analysis (page 7, lines 173 – 180), since we are looking for trends and an overview rather than pinpointing specific genes, we chose a more liberal approach by pooling all the genes from any of the four methods. In fact, we only pooled the genes together for the pathway enrichment analysis. For the “Drug for AAA with comorbid conditions”, we utilized the disease genes strictly defined by the four-method strategy. We have revised the manuscript to explain how the genes are selected. Furthermore, we updated the interpretation results of shared SNVs around lipid genes by correcting on the counts of trait pairs and focusing mostly on the shared causal SNVs:

page 6, lines 152 – 153: “We observed the shared SNVs, both causal and non-casual, clustered proximal to lipid-related genes (Supplementary Table 4).”

page 6-7, lines 153 – 157: “For example, *LPA* was the closest gene for 9 SNVs shared by AAA and 11 other traits, among which rs10455872²⁹ was causal to 4 trait pairs, and rs140570886³⁰, rs76735376, and rs6905073 were shared by at least 3 trait pairs. Similarly, *CDKN2B-AS1* was annotated to 8 shared SNVs shared by 10 trait pairs, including rs1537371³¹ which was causal to 3 trait pairs.”

page 7, lines 162-166: “Annotating GWAS variants to genes solely by proximity is oversimplified and may not account for pleiotropy. We therefore adopted four approaches, TWAS-Fusion³⁹, SMR⁴⁰, MAGMA⁴¹, and GCTA⁴² to infer shared genes (Supplementary Fig. 3). Among

these methods, the first two leverage expression quantitative trait loci (eQTL), and the latter two mainly utilize proximity for gene burden tests.”

page 7, line 173: “Pooling genes from any of the four methods for an overview of biological pathways, ...”

Indeed, we utilized the pre-computed models in GTEx v6 for TWAS-Fusion for identifying GWAS genes, which is now described in our method section (Page 16, lines 480-483). For SMR, we chose to restrict our analysis to 9 tissues, because the associations in the causal tissue were consistently stronger than those in the non-causal tissue (ref 4). Further, given that SMR is computationally intensive, we curated a list of AAA-relevant tissues based on the tissue-specific enrichment of AAA GWAS summary stats (single trait GWAS). We also supplemented the list with tissues heavily involved in cardiovascular disease, such as heart, kidney, and whole blood. Although only representative cardiovascular tissues were included in our study, the result provides ample information for explaining biological mechanisms.

Page 16, lines 481-483: “We conducted TWAS using the FUSION software³⁸ based on 43 post-mortem tissue expression profiles in GTEx (version 6, with pre-computed models)⁴⁴.”

REFERENCE:

1. Roychowdhury T, Klarin D, Levin M G, et al. Genome-wide association meta-analysis identifies risk loci for abdominal aortic aneurysm and highlights PCSK9 as a therapeutic target[J]. Nature genetics, 2023, 55(11): 1831-1842.
2. Gazal S, Weissbrod O, Hormozdiari F, et al. Combining SNP-to-gene linking strategies to identify disease genes and assess disease omnigenicity[J]. Nature genetics, 2022, 54(6): 827-836.
3. Aragam K G, Jiang T, Goel A, et al. Discovery and systematic characterization of risk variants and genes for coronary artery disease in over a million participants[J]. Nature Genetics, 2022, 54(12): 1803-1815.
4. Hu Y, et al. A statistical framework for cross-tissue transcriptome-wide association analysis. Nat Genet. 2019 Mar;51(3):568-576.

2.2) They used PLINK clumping to find independent SNVs for the functional annotation, then FM-summary to find a credible set for fine mapping. Why two methods to produce the same type of output?

Response:

The outputs of these two methods are different. Clumping is used to find independent SNVs per region of LD, after which representative SNVs with the smallest P values are retained and other SNVs in LD are pruned. However, these independent SNVs are not necessarily causal, so therefore we adopted fine-mapping to further infer causal variants shared by AAA and other traits. In order to clarify this point, in the revised manuscript, we deleted the table of fine-mapped 99% credible SNVs, i.e., Supplementary Table 4, and only present the final result of shared causal variants (previous Supplementary Table 5, and now the new Supplementary Table 4).

3) Please explain the following sentence (line 143): “Repressors, enhancers and promoters tend to have stronger than the genome-wide correlations (Supplementary Fig. 1), suggesting transcription regulation is genetically shared.”. The authors assumed the

reader is fully familiar with LDSC and that “stronger than the GW correlations” has a full meaning. It does not. It is not clear how the authors reach the conclusion that transcription regulation is shared (across traits?).

Response:

Thank you for your suggestion. We meant to emphasize that these regulatory elements were correlated the strongest. In the revised manuscript, we rewrote this to explain the results more directly:

Page 5-6, lines 124-126: “Repressors, enhancers and promoters tend to have the strongest correlations across traits (Supplementary Fig. 1), suggesting transcriptional regulation is genetically shared.”

4) Lines 143-145: “Among the 47 shared causal variants, only four were local lead SNVs (Fig. 3B), i.e., having the smallest GWAS P values, whereas the rest were located near the lead SNVs, suggesting the importance of fine-mapping.”. What are the authors trying to tell us here? Fine mapping methods were developed to identify causal variants resolving issues such as LD contamination, it is not the result of their analyses.

Response:

We meant to stress that causal variants do not necessarily have the smallest P values in association tests. As this sentence causes confusion, we revised it to:

Page 6, lines 149-151: “Among the 47 shared causal variants, only four had the smallest GWAS P values (Fig. 3B), reinforcing that local lead SNVs in GWAS may only tag the causal SNVs²⁶.”

5) Line 281: “[...] hypertension, which is a disease rather than a physiological trait.”. This sentence makes no sense. Diseases and traits are all the results of genetic and environmental factors, so I am not sure why the authors want to make a distinction here that have no added value.

Response:

We thank the reviewer for the comment. Diseases present severe forms of body malfunction, however, it's indeed hard to justify that diseases should be more closely related to each other than a disease with a physiological trait. The original statement is therefore confusing. However, hypertension is an established risk factor for AAA.

Placing it as a yardstick helps to understand the close relationship between lipid traits and AAA. As such, we revised it to:

Page 10, lines 262-263: “First, lipid traits rank as the second strongest causal factor for AAA (OR = 1.46-1.73), next to hypertension, an established risk factor for AAA.”

6) The authors are assuming some kind of directionality in some of their results. For example, line 291: “These results reinforce the notion that predisposition to lipid malfunction is a strong feature in CMTs” The results do not indicate directionality. There is no information that would be derived about whether lipids are malfunctioning (?) or not in relation to CMTs. If there is some form of directionality, it needs to be made clear.

Response:

We thank the reviewer for the comment. We should have made it clear that the directionality comes from MR, TWAS and SMR analyses. We have revised it to the following:

Page 10, lines 265-267 “Third, the most broadly shared genes between AAA and CMTs, i.e., *LRP1*, *PSRC1*, *CELSR2*, and *NOC3L*, are all lipid related and their up or down regulation was associated with CMTs (Supplementary Fig. 8).”

REVIEWER #3

1) What are the noteworthy results?

This review is a comprehensive overview of the genetic findings that overlap between cardiovascular diseases and aneurysm disease. The findings do of course by the nature of being a review correspond well with the reports in the field, and maybe do not share great new findings, but on the other hand clarify and enlighten important shared areas within genetics for these patient groups.

Response:

We thank the reviewer for appreciating our effort of a systematic interrogation of AAA and cardiometabolic traits from the genetics perspective.

2) Will the work be of significance to the field and related fields?

This review is comprehensive and potentially can be important and contribute to future lines to follow especially in the AAA area.

Response:

We thank the reviewer for the positive comment.

3) Does the work support the conclusions and claims, or is additional evidence needed? It could be a more broad inclusion of more relevant clinical papers supporting for example the influence by lipid pathology or statin therapy as support of the findings. There is a general lack of clinical aspect. An example could be Line 267-268; many clinical papers also support this. Please include and thereby the clinical relevance of the findings will be upgraded!

Response:

We thank the reviewer for the comment. Indeed, interpreting our results with a clinical perspective will add value to our study. In our revised manuscript, we cited a few clinical studies on AAA and CAD (Page 9-10, lines 250-251) and clinical trials of statin therapy for treating AAA (Page 10, lines 273-275). Furthermore, we added two supplementary tables (6 and 7) for an overview of the trails on candidate drugs identified in our study (Page 9, lines 221-222, lines 236-238; Page 12, lines 331-335).

Page 9-10, lines 250-251: “Epidemiological studies reported many risk factors common to AAA and CAD⁵⁵, and the two diseases tend to co-occur^{56,57}.”

Page 10, lines 273-275: “Our study also supports a higher burden of inherited dyslipidaemia in patients of AAA, and lowering LDL-C serves as a therapeutic strategy for preventing and managing AAA^{59, 60, 61, 62}”

Page 9, lines 221-222: “Most of these drugs have been approved to treat various cardiovascular diseases (Supplementary Table 6).”

Page 9, lines 236-238: “Most of these herb products are in phase 3 clinical trials (Supplementary Table 6) and have shown potential in preventing and treating CVDs, including AAA^{51, 52, 53, 54}.”

Page 12, lines 331-335: “Most drug candidates we discovered have been used to treat CVDs, and some are in clinical trials for repurposing to treat AAA, including atorvastatin, curcumin, metformin, and five other drugs (Supplementary Table 7). Note that numerous drugs with high matching scores in our analysis are not in clinical tests, inviting future studies for their therapeutic potential.”

4) Are there any flaws in the data analysis, interpretation and conclusions? Do these prohibit publication or require revision?
no apparent for me.

Response:

We thank the reviewer for the evaluation.

5) Is the methodology sound? Does the work meet the expected standards in your field?
Yes, this appears well performed, and cover and include all the more recent published relevant work in the field.

There is possibly a lack of epigenetic reflections; we know as clinicians the profound influence by smoking on disease development and expansion for aneurysms, could this be attributed some comments/other analysis?

Response:

We thank the reviewer for the suggestion. Although epigenetics is out of our study scope, it is a subject of focus in AAA biology. In the revised manuscript, we refer to it in the Discussion section:

Page 12, lines 340-343: “Second, we focus on genetics in this study as CMTs harbor high heritability in general, however, other factors such as epigenetics can also play important roles. As an example, smoking, as an established risk factor of AAA, is known to cause vast epigenetic changes⁸⁶.”

6) Is there enough detail provided in the methods for the work to be reproduced?

Specific review comments:

The text which refers to more clinical data or are disease-related are sometimes not correct or clear.

Abstract: "AAA presents abnormal metabolism – would never be used in a clinical context reporting on AAA, what do the authors mean?"

Response:

We thank the reviewer for pointing it out. We have revised it in the abstract:
Page 2, lines 23-24: “Abdominal aortic aneurysm (AAA) has a high heritability and often co-occurs with other cardiometabolic disorders, suggesting shared genetic susceptibility.”

7) Lack SNV in abbreviation list.

Response:

Added in the revised manuscript. We thank the reviewer for pointing it out.
Page 3, line 55: “SNV: single nucleotide variants”

8) Line 81-82 The authors could benefit from more updating to more contemporary references for AAA disease. The prevalence rate is rarely 3-9%, even if it refers to a recent review in EHJ, there are many more contemporary prevalence reports rather going down to 0.5(in women) to 4% in men. Please revise and review.

Response:

We thank the reviewer for the constructive comment! In our revised manuscript, we updated the epidemiologic literature related to AAA disease:

Page 4, line 64-66: “Abdominal aortic aneurysm (AAA), defined as focal dilation of the abdominal aorta by 50% or reaching ≥ 30 mm in diameter, is a complex vascular disease with an estimated global prevalence of 0.92%¹.”

9) Line 84. Question this report, there is much stronger evidence for rupture rate in other papers. Please revise. (3)

Response:

We thank the reviewer for the constructive comment. We checked literature more closely and updated the AAA rupture rate in the revised manuscript:

Page 4, line 67-68: “Once reaching 55 mm, the five-year cumulative rupture rate is 25-40%².”

10) Line 84-85 This paper (4) is also a bit odd as it refers to a review but reports it as one number please revise and maybe add more international and clear report-data?

Response:

We thank the reviewer for the constructive comment. We have revised our citation:

Page 4 line 68-69: “Among ruptured patients, a mortality rate as high as 80% was observed³, rendering AAA a leading cause of death. ”

Ref 3: J J Reimerink, M J van der Laan, M J Koelemay, R Balm, D A Legemate, Systematic review and meta-analysis of population-based mortality from ruptured abdominal aortic aneurysm, *British Journal of Surgery*, Volume 100, Issue 11, October 2013, Pages 1405–1413.

11) Line 86: Pathologically is this really a good choice of word here? The pathogenesis is...?

Response:

We thank the reviewer for the comment. We revised it to:

Page 4 line 70-71: “AAA is characterized by remodeling and degradation of extracellular matrix, apoptosis of smooth muscle cells, luminal thrombosis, and chronic inflammation ^{4, 5}.”

12) Section line 86-100 Please look into your choice of reporting the literature. The reader gets the impression that you have a certified causality between hyperlipidemia and AAA. It is also strange to read that line 92-95 that one has hypercholesterolemia in AAA patients, as if this a certified association in all patients, which it is not. Please revise.

Response:

We thank the reviewer for the constructive comment. Indeed, not all AAA patients would have hyperlipidemia. We added words “can be” and “Often” to make it sound less decisive: Page 4, lines 74-79: “Meanwhile, metabolic homeostasis **can be** perturbed, resulting in enhanced glycolysis in the aortic wall ⁷ and altered serum levels of amino acids and lipids ^{8, 9, 10}. **Often**, circulating total cholesterol, low-density lipoprotein cholesterol (LDL-C), triglycerides, and sulfur amino acids are elevated, whereas high-density lipoprotein cholesterol (HDL-C) and phosphatidylcholines are reduced.”

13) Line 272 "... consistent with their common features of artery malfunction and atherosclerosis" really unclear of the meaning of this? Other/more traits should then be included? Please revise and explain?

Response:

We thank the reviewer for pointing this out. We revised the sentence to be more specific of the tissue functions:

Page 10, lines 252-254: “These shared signals were enriched for artery and liver tissues, reflecting their common malfunctions in artery and lipid metabolism.”.

14) rad 289-290 Unsure of the meaning of this kind of text: " were suggested to treat many AAA with comorbid disease", please revise and change.

Response:

We thank the reviewer for pointing this out. We revised the sentence to:

Page 10, line 271-272: “lipid reducing drugs were suggested as strong candidates to treat AAA with various comorbid CMTs”.

15) rad 291 "In comparison, glucose traits demonstrate neither correlation of causality to AAA" -- perhaps some discussion about this in comparison to the literature?

Response:

This was raised by another reviewer too! Indeed, this is a common interest in the field of AAA biology. We added a short discussion of this observation in the revised manuscript: Page 10, lines 276-281: “In comparison, glucose traits demonstrate neither genetic correlation nor causality to AAA. The relationship between glucose and AAA have been paradoxical. While inverse correlation are reported for the risk and growth rate of AAA and the HbA1c level ⁶², fasting glucose level ⁶³, and T2D ⁶⁴, no genetic correlation has been reported thus far ⁶⁵, inviting further investigations. ”

16) Last section; line 555-567 Very interesting and relevant analysis. However there is a lack of reflection on these drugs as compared to already performed or planned RCT trials in the field. Please add the corresponding trials if there are some in this list? Such as statins etc and correlate to your suggestions?

Response:

We thank the reviewer for appreciating our effort in trying to find therapeutic targets and drugs for treating AAA in the presence of comorbid diseases. Following the advice, we located relevant clinical trials of the drug candidates identified in our analysis, and found some are being tested for treating AAA, indeed. We revised our manuscript accordingly (partially overlap with Question 3):

Page 9, lines 221-222: “Most of these drugs have been approved to treat various cardiovascular diseases (Supplementary Table 6).”

Page 9, lines 236-238: “Most of these herb products are in phase 3 clinical trials (Supplementary Table 6) and have shown potential in preventing and treating CVDs, including AAA^{51, 52, 53, 54}.”

Page 12, lines 331-335: “Most drug candidates we discovered have been used to treat CVDs, and some are in clinical trials for repurposing to treat AAA, including atorvastatin, curcumin, metformin, and five other drugs (Supplementary Table 7). Note that numerous drugs with high matching scores in our analysis are not in clinical tests, inviting future studies for their therapeutic potential.”

REVIEWERS' COMMENTS

Reviewer #1 (Remarks to the Author):

The authors have answered the concerns to the extent the current knowledge allows, so acceptance of the manuscript is recommended. In fact, it is extremely interesting the fact of the inverse correlation between T2D and AAA in epidemiological studies, with no genetic correlation observed. Out of scope for the present manuscript, but would want to add a suggestion for future research. As it is well known that T2D has two main different components, the insulin resistance part and the pancreatic-insulin production/secretion part, it may be interesting to analyze the association with AAA separately, if possible. Maybe one way to start would be to take a look to the genes/variants for the different T2D genetic subgroups defined by Udler et al.

Reviewer #2 (Remarks to the Author):

I feel my comments and point we appropriately answered.

Reviewer #3 (Remarks to the Author):

The authors have meet the critics from us and others

Thank you. No further major comments.